# A Sulfuric Acid Nucleation Potential Model for the Atmosphere

Jack Johnson[1,2] and Coty N. Jen[1,2]

1. Chemical Engineering, Carnegie Mellon University, Pittsburgh, PA, 15213, USA

2. Center for Atmospheric Particle Studies, Carnegie Mellon University, Pittsburgh, PA, 15213, USA

*Correspondence to:* Coty N. Jen (cotyj@andrew.cmu.edu)

**Abstract.** Observations over the last decade have demonstrated that the atmosphere contains potentially hundreds of compounds that can react with sulfuric acid to nucleate stable aerosol particles. Consequently, modeling atmospheric nucleation requires detailed knowledge of nucleation reaction kinetics and spatially and temporally resolved measurements of

numerous precursor compounds. This study introduces the Nucleation Potential Model (NPM), a novel nucleation model that dramatically simplifies the diverse reactions between sulfuric acid and any combination of precursor gases. NPM predicts 1-nm nucleation rates from only two measurable gas concentrations, regardless of whether all precursor gases are known. NPM describes sulfuric acid nucleating with a parameterized base compound at an effective base concentration, $[B_{eff}]$. $[B_{eff}]$ captures the ability of a compound or mixture to form stable clusters with sulfuric acid and is estimated from measured 1-nm particle

concentrations. NPM is applied to experimental and field observations of sulfuric acid nucleation to demonstrate how $[B_{eff}]$ varies for different stabilizing compounds, mixtures, and sampling locations. Analysis of previous field observations shows distinct differences in $[B_{eff}]$ between locations that follow the emission sources and stabilizing compound concentrations for that region. Overall, NPM allows researchers to easily model nucleation across diverse environments and estimate the concentration of non-sulfuric acid precursors using a condensation particle counter.

## 1 Introduction

Atmospheric aerosol particles play an important role in cloud formation and Earth's radiation balance. Global climate models estimate that around 50% of cloud condensation nuclei (CCN) are produced by nucleation (Gordon et al., 2017; Yu and Luo, 2009; Merikanto et al., 2009; Spracklen et al., 2008), whereby gas-phase compounds react and form a stable particle approximately 1-nm in diameter (Jen et al., 2015; Chen et al., 2012). As a result, nucleation influences cloud properties and

lifetimes, which subsequently impact Earth's radiation balance (Spracklen et al., 2008, 2006). Therefore, accurate modeling of nucleation rates in the atmosphere is necessary to predict atmospheric aerosol concentrations used in global weather and climate models.

Aerosol nucleation in the troposphere is primarily driven by sulfuric acid (Kuang et al., 2008; Sihto et al., 2006; Sipilä et al., 2010; Lee et al., 2019; Weber et al., 1996, 1997; Kulmala et al., 2004) which reacts with a large variety of compounds,

to form particles (Kürten et al., 2016a; Glasoe et al., 2015; Weber et al., 1998; Kirkby et al., 2011; Jen et al., 2014; Coffman and Hegg, 1995; Almeida et al., 2013). Laboratory studies have demonstrated that sulfuric acid nucleates with various compounds at rates spanning over seven orders of magnitude (Elm et al., 2016; Jen et al., 2016, 2014; Kürten et al., 2014; Glasoe et al., 2015). The ever-expanding list of compounds includes ammonia (Kirkby et al., 2011; Hanson and Eisele, 2002;

Coffman and Hegg, 1995), amines (Glasoe et al., 2015; Kurtén et al., 2008; Jen et al., 2014), diamines (Elm et al., 2016; Jen et al., 2016; Elm et al., 2017), alcohol amines (Xie et al., 2017), organic acids (Zhao et al., 2009; Zhang et al., 2004), oxidized organics (Riccobono et al., 2012, 2014; Ehn et al., 2014; Zhao et al., 2013), water (Kulmala et al., 1998; Merikanto et al., 2007), and ions (Eisele et al., 2006; Kirkby et al., 2011). Additionally, sulfuric acid has been shown to nucleate with multiple compounds synergistically, such as dimethylamine/ammonia (Glasoe et al., 2015; Yu et al., 2012) and oxidized organics/ammonia (Lehtipalo et al., 2018).

Currently, three classes of nucleation models are used to estimate atmospheric nucleation rates, but no existing model is capable of capturing the true complexity of atmospheric nucleation reactions. First, power-law nucleation models estimate nucleation rates from empirically derived power-law functions fitted from measured nucleation rates with concentrations of sulfuric acid with various precursor gases (Yao et al., 2018; Glasoe et al., 2015; Kirkby et al., 2011). These power-law models have been used to predict nucleation rates in areas such as Asian megacities, the Amazon Rainforest, and globally (Yao et al., 2018; Zhao et al., 2020; Dunne et al., 2016). The fitted coefficient and exponentials on the precursor concentration may be indicative of key rate-limiting steps (Sihto et al., 2006) or may have no physical meaning (Kupiainen-Määttä et al., 2014). Furthermore, The power-law models are typically only dependent on two to three nucleation precursor concentrations, and thus cannot accurately predict nucleation rates in areas where numerous and unknown compounds are nucleating with sulfuric acid (Zhao et al., 2020). Computational chemistry nucleation models compute formation free energies of clusters containing sulfuric acid and stabilizing compounds in order to numerically solve the cluster balance equations (Ortega et al., 2012; Myllys et al., 2018; McGrath et al., 2012; Olenius et al., 2013; Elm, 2019; Yu et al., 2018). While computational chemistry models can rigorously show the formation pathways of sulfuric acid clusters, the method becomes too computationally expensive when determining formation pathways for a mixture of nucleating compounds. Finally, acid-base nucleation models are based on experimentally observed nucleation kinetics that have demonstrated particles form via the sequential addition of acid and base molecules (Chen et al., 2012; Jen et al., 2014; Kürten et al., 2018). These experiments use a chemical ionization mass spectrometer (CIMS) to measure gas and cluster concentrations to estimate cluster evaporation rates. Though acid-base models can experimentally determine the reaction kinetics of sulfuric acid clusters, finding evaporation rates for numerous cluster types is experimentally arduous due to its dependence on nucleation precursor composition and concentration. While each model type provides unique and beneficial information about how sulfuric acid nucleates, they fail to predict particle nucleation rates in complex mixtures, such as the atmosphere, and require high spatial and temporal speciated precursor measurements to accurately predict global nucleation rates.

Currently, most global climate models only account for sulfuric acid binary or ternary nucleation with water or water and ammonia (Semeniuk and Dastoor, 2018). Only a few models incorporate power-law nucleation models (Gordon et al., 2017; Zhao et al., 2020; Dunne et al., 2016). However, experimental observations indicate that even low concentrations of other stabilizing compounds can enhance sulfuric acid nucleation rates beyond those predicted from models (Li et al., 2020; Wang et al., 2018). Moreover, many emission inventories used in global climate models only contain emission factors for sulfur dioxide and ammonia (Semeniuk and Dastoor, 2018; Lee et al., 2013; Dunne et al., 2016; Spracklen et al., 2008) with

some including volatile organic compounds (Hoesly et al., 2018). Furthermore, only sparse measurements, both in time and space, exist of the numerous precursor compounds in the atmosphere. Combined, these factors contribute to significant model

error in predicting aerosol number concentrations in regions with no dominant nucleation pathway (Dunne et al., 2016; Kerminen et al., 2018; Ranjithkumar et al., 2021).

This study presents a generalized, semi-empirical model for sulfuric acid nucleation, known as the Nucleation Potential Model (NPM), that simplifies the numerous and often unknown nucleation reactions into a single reaction pathway. Specifically, NPM reflects how sulfuric acid reacts with an effective base compound and predicts 1-nm nucleation rates from

sulfuric acid and a parameterized base concentration ($[B_{eff}]$). $[B_{eff}]$ captures the combined concentrations of compounds and their ability to stabilize sulfuric acid clusters. This parameterized concentration is estimated from measured 1-nm particle concentrations formed from controlled reactions between sulfuric acid and a complex mixture. This study demonstrates the dependencies of $[B_{eff}]$ from a variety of stabilizing gas mixtures and how $[B_{eff}]$ varies across diverse regions of the world.

The full impact of using the Nucleation Potential Model is two-fold: (1) The effective nucleation precursor

concentration needed to predict 1-nm nucleation rates can be measured with a portable and cost-effective condensation particle counter (CPC), instead of a mass spectrometer. The increased development and deployment of 1-nm CPCs (Hering et al., 2017; Lehtipalo et al., 2022; Kuang, 2018) will enable researchers to measure $[B_{eff}]$ at high spatial and temporal resolution which is currently challenging to achieve with mass spectrometers. Furthermore, the combined observations from NPM with a CPC and mass spectrometry will also provide a detailed understanding on which compounds nucleate and the rate at which they

nucleate. In addition, (2) the NPM is currently the only model that can represent nucleation of arbitrarily complex mixtures of compounds found in the atmosphere.

## 2 Methodology

### 2.1 Model Description

The Nucleation Potential Model (NPM) generalizes the formation of 1-nm particles from sulfuric acid nucleation as

a series of second-order reactions. Reaction 1 shows the reaction pathway for the NPM, where *n* represents the number of sulfuric acid (*A*) and base (*B*) molecules in a cluster. $N_n$ denotes the cluster size with $N_1$ as the monomer (i.e., one sulfuric acid molecule with that same number of base or other attached compounds) up to $N_4$ as the tetramer. The reaction pathway is based on the most energetically probable pathway for sulfuric acid and base clusters to form, with less probable pathways excluded to reduce model calculation time and complexity (Olenius et al., 2017). The final step in Reaction 1 is the formation of the

tetramer, $N_4$. At the tetramer size, the particles are approximately 1-nm in diameter or 1.3-nm in mobility diameter (Chen et al., 2012; Jen et al., 2015; Larriba et al., 2011). Coagulation losses are estimated from the collision rate constant between clusters. Any cluster formed through $N_8$ in size is accounted for in the total concentration of particles. Coagulation loss to larger particles (i.e., growth to sizes larger than $N_8$) is not included in this model when no pre-existing particles are present.

Coagulation to pre-existing particles is included as a separate loss term when analyzing ambient observations. Cluster balance

equations (i.e., rates laws) for Reaction 1 are provided in the supplementary information (SI, Equation S1).

$$A_1 + B_{eff} \xrightarrow{k} A_1 \cdot B_{eff}$$

$$N_n = A_n \cdot B_n$$

$$N_1 + N_1 \xrightarrow{k} N_2$$     **Reaction 1**

$$N_1 + N_2 \xrightarrow{k} N_3$$

$$N_2 + N_2 \xrightarrow{k} N_4$$

$$N_1 + N_3 \xrightarrow{k} N_4$$

The forward reaction constant is assumed to be equal for all clusters at $k = 4.2 \times 10^{-10}$ cm$^3$ s$^{-1}$ and is the collision rate constant calculated using parameters estimated from density functional theory and bulk properties (Ortega et al., 2012). The effective base concentration ([$B_{eff}$]) represents the stabilization effects that a compound or mixture of compounds has on the formation rate of sulfuric acid clusters. [$B_{eff}$] also depends on the nucleation precursors' concentrations, composition,

temperature, and humidity. A compound that effectively stabilizes sulfuric acid clusters has a higher value for [$B_{eff}$] than a weaker stabilizing compound. [$B_{eff}$] is numerically solved from the cluster balance equations (Equation S1) with inputs of the initial concentration of sulfuric acid monomer ([$A_1$]$_o$), the final concentration of nucleated 1-nm particles (i.e., [$N_4$]), and nucleation reaction time ($t_{nucl.}$).

## 2.2 Experimental Setup

[$B_{eff}$] was determined for nucleating systems consisting of sulfuric acid and various combinations of atmospherically relevant bases reacting in an extremely clean and repeatable flow reactor at 300 K and 20% relative humidity (RH). [$B_{eff}$] is likely influenced by temperature and RH. Lowering temperature would stabilize sulfuric acid clusters, leading to an increase in [$B_{eff}$] (Hanson and Eisele, 2002; Vehkamäki et al., 2002; Dunne et al., 2016). The effects of RH are not clear and would depend on the concentration and composition of the other nucleation precursor vapors in the system (Olenius et al., 2017; Ball

et al., 1999; Henschel et al., 2014; Merikanto et al., 2007). Future experiments will examine NPM over a wider range of temperature and RH to determine the impact this has on [$B_{eff}$]. The flow reactor system used for these measurements was constantly purged with a mixture of sulfuric acid, nitrogen, and water (Fomete et al., 2021; Ball et al., 1999; Jen et al., 2014). This creates extremely clean and repeatable conditions in the reactor. Baseline measurements are taken daily to verify the flow reactor's cleanliness and repeatability in concentration, temperature, and RH. The method for these baseline measurements is

described in Fomete et al., 2021. [$A_1$]$_o$ and base concentrations ([B]) were measured with a custom-built, transverse atmospheric pressure acetate/hydronium chemical ionization inlet coupled to a long time-of-flight mass spectrometer

(Pittsburgh Cluster CIMS, PCC) (Fomete et al., 2021). The bases included dilute concentrations of ammonia ($NH_3$), methylamine (MA, $CH_3NH_2$), dimethylamine (DMA, $(CH_3)_2NH$), and trimethylamine (TMA, $C_3H_9N$) that are injected into the flow reactor by flowing nitrogen over a custom-made permeation tube (Fomete et al., 2021; Zollner et al., 2012). The $t_{nucl}$ was determined to be 2 s from the modeled centerline velocity of the reactor (Hanson et al., 2017; Panta et al., 2012). The concentrations of $N_4$ and larger particles were measured with a 1-nm versatile water-based Condensation Particle Counter (vwCPC, TSI 3789) (Hering et al., 2017). The flow tube was optimized to minimize the concentration of particles >1-nm by lowering the sulfuric acid monomer concentration ($[A_1]_o$). This was done to prevent the vwCPC from saturating and minimize particle coagulation with particles larger than $N_8$. See Figure S1 for more details on 1-nm particle optimization experiments.

## 3 Results and Discussion

### 3.1 Experimental Model Validation

Figure 1 shows $[B_{eff}]$ for the single component injections of $NH_3$, MA, DMA, and TMA in the sulfuric acid reactor. These atmospherically relevant compounds have previously been shown to nucleate with sulfuric acid at different rates (Jen et al., 2016, 2014; Kurtén et al., 2008; Glasoe et al., 2015). $[A_1]_o$, was measured daily and ranged between $9x10^7 \, cm^{-3}$ to $3x10^8 \, cm^{-3}$. Daily measurements of $[A_1]_o$ were then used as the initial concentration of sulfuric acid in the NPM. The average value for $[A_1]_o$ ($[A_1]_{o,avg}=1x10^8 \, cm^{-3}$) will be used for simplicity. While $[A_1]_o$ is higher than those typically measured in the atmosphere, any range of $[A_1]_o$ can be modeled as this parameter is an input to the NPM. Each base compound was injected at various measured [B], ranging from 0.5 to 32 pptv. The base concentrations examined in this study fall within the range observed in the atmosphere (Hanson et al., 2011; Cai et al., 2021; Kürten et al., 2016b). Note, the error bars in Fig. 1 represent how the standard deviation in particle concentration measurements effects $[B_{eff}]$.

From Fig. 1, $[B_{eff}]$ for $NH_3$ remains unchanged at approximately 10-15 pptv across the entire $[NH_3]$ range. This constant $[B_{eff}]$ trend suggests that $NH_3$ does not significantly stabilize sulfuric acid clusters and enhance nucleation rates under the experimental conditions in the flow tube. This is expected due to the relatively short nucleation time when compared to previous flow reactor studies (Jen et al., 2016; Glasoe et al., 2015). In contrast, $[B_{eff}]$ increases up to ~40 pptv with increasing [MA], demonstrating that this compound enhances sulfuric acid nucleation more than $NH_3$. The $[B_{eff}]$ curves for DMA and TMA exhibit higher slopes than MA and $NH_3$, indicating that DMA and TMA substantially enhance sulfuric acid nucleation rates at low [B]. Furthermore, at [B] = 10 pptv, $[B_{eff}]$ for DMA and TMA are two to three times higher than MA and four to six times higher than $NH_3$. This indicates that DMA and TMA have a much stronger interaction with sulfuric acid clusters than MA and $NH_3$. Note, the plateau in $[B_{eff}]$ occurs when a significant concentration of >1-nm particles at high [B] increases the coagulation rate beyond what is predicted by the NPM (up to $N_8$). The relative strength of these compounds in enhancing nucleation is consistent with previously published results indicating that the NPM is correctly capturing the nucleation potency of $NH_3$, MA, DMA, and TMA (Glasoe et al., 2015; Jen et al., 2014; Kürten et al., 2018).

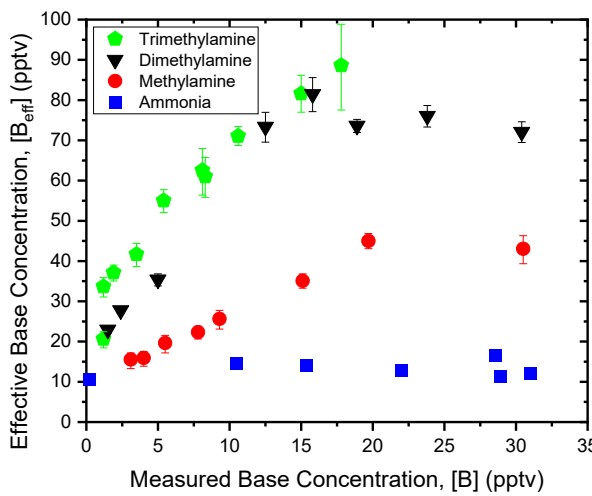

**Figure 1:** Comparison of effective base concentration from NPM ([$B_{eff}$]) with measured base concentration ([B]) for single component injections of ammonia (blue squares), methylamine (red circles), dimethylamine (black triangles), and trimethylamine (green pentagons). The average sulfuric acid concentration was $1 \times 10^8$ cm$^{-3}$, and the reaction time was 2 s.

NPM was also used to determine [$B_{eff}$] for more complex mixtures of nucleation precursors. Figure 2 shows [$B_{eff}$] from simultaneous injections of NH$_3$ at 73 pptv and varying [DMA] into the sulfuric acid flow reactor. NH$_3$ and DMA mixture injections have higher values for [$B_{eff}$], up to 120 pptv, which are especially prominent at higher concentrations of DMA. At [B] = 20 pptv, [$B_{eff}$] for the mixture of NH$_3$ and DMA is significantly higher than linear addition of the [$B_{eff}$] from individual DMA and NH$_3$, ~110 pptv compared to ~80 pptv, respectively. This suggests that DMA and NH$_3$ react synergistically with

sulfuric acid to form particles. The synergist effect is due to ammonia's ability to stabilize sulfuric acid clusters long enough for DMA to collide and react with the sulfuric acid-ammonia clusters (Myllys et al., 2019; DePalma et al., 2012; Glasoe et al., 2015).

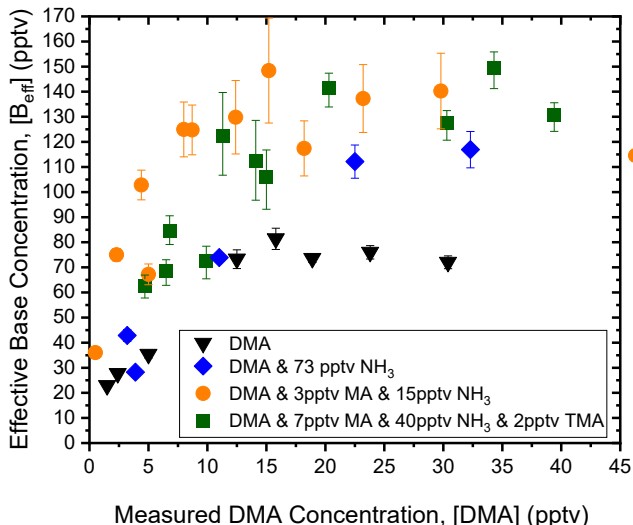

**Figure 2:** Comparison of [$B_{eff}$] and measured dimethylamine (DMA) concentration for multi-component injections. Mixture experiments for DMA (black triangles), DMA with 73 pptv NH$_3$ (blue diamonds), DMA with 7 pptv MA, 40 pptv NH$_3$ and 2 pptv TMA (green squares), and DMA with 3 pptv MA and 15 pptv NH$_3$ (orange circles). The average sulfuric acid concentration was $1 \times 10^8$ cm$^{-3}$ and a reaction time of 2 s.

165          Figure 2 also shows mixtures containing combinations of $NH_3$, MA, and TMA with varying amounts of DMA. Again, an increase in [DMA] leads to an increase in [$B_{eff}$], and all mixture curves display an enhancement to nucleation compared to pure sulfuric acid-DMA nucleation. There is no significant distinction in the trends of [$B_{eff}$] between the 3 and 4 component mixture curves. This could be due to the higher base concentrations in these systems compared to the sulfuric acid concentration which results in particles being formed at the sulfuric acid collision limit. Addition of more bases could also

help grow particles, causing higher coagulation losses not captured in the coagulation loss term in NPM. As discussed further in the SI, NPM only accounts for coagulation with particles up to of $N_8$ in size, indicating that NPM may not be capturing coagulation effects in the system saturated with base. Additionally, [$B_{eff}$] is ~ 60 pptv at 10 pptv of DMA in the DMA and $NH_3$ curve in Fig. 2, while [$B_{eff}$] is ~100 pptv at 10 pptv of DMA for the 3 and 4 component mixture curves in Fig. 2. These observations imply that $NH_3$ and DMA are reacting synergistically with sulfuric acid, while MA and TMA are individually

reacting with sulfuric acid to contribute the additional 40 pptv to [$B_{eff}$]. However, a computational chemistry model is required to draw further conclusions on how these molecules are reacting in a complex mixture. Regardless, observations from Fig. 2 indicate that NPM determines to what extent a complex mixture of compounds will enhance sulfuric acid nucleation solely using measurements from the vwCPC.

          The measured uncertainty in [$B_{eff}$] observed for the mixture experiments in Fig. 2 is higher than the single-component

results (Fig. 1). The error bars were estimated from the standard deviation in the concentration of particles for each experiment. Fluctuations in particle concentrations capture the small variation in injected base concentrations, as well as disruption to the flow profiles. Additionally, the mixture experiments were measured over multiple days while many of the single component measurements were taken in 1-2 days. There are likely small day-to-day changes in the mixing within the dilution system which would increase the uncertainty across a longer time frame of measurements. The overall uncertainty in [$B_{eff}$] is also

primarily influenced by the uncertainty in the particle size distribution, and to a lesser extent the particle concentration measurements, measured concentrations of gas-phase compounds, the flow dynamics within the flow reactor, temperature, and humidity. The estimated systematic uncertainty in PCC measurement of [$A_1$]$_o$ and [B] are approximately a factor a two and would not impact the trends observed in Figures 1 and 2 (Zhao et al., 2010; Simon et al., 2016; Erupe et al., 2010). Currently, daily baseline measurements were taken following the procedure in Fomete et al. (2021) to ensure consistent and stable

concentrations of both gas-phase and particle-phase compounds within the flow reactor. Furthermore, the measured particle concentrations are not corrected for detection efficiency as it is not known for electrically neutral sulfuric acid-amine 1-nm particles. The detection efficiency of clusters composed of sulfuric acid and amines/ammonia is normally assumed to be similar, and thus accounting for this will not impact the reported [$B_{eff}$]. In future studies, electrically neutral size distributions will be measured to constrain the coagulation rates in NPM.

## 3.2 Estimation of [$B_{eff}$] in Various Regions of the World

The NPM was also used to determine how the effective concentration of stabilizing compounds vary around the world. Nucleation rates of 1-nm particles ($J_{1nm}$, which equals the formation rate of $N_4$) and sulfuric acid concentrations were obtained from previous field campaigns including in Hyytiälä Forest, Finland (Sihto et al., 2006); Mexico City, Mexico (Iida et al., 2008); Atlanta, Georgia (McMurry and Eisele, 2005); Boulder, Colorado (Eisele et al., 2006); and Beijing, China (Cai et al., 2021). The equations of the NPM (Equation S1) were solved at steady state to determine [$B_{eff}$] from the observed $J_{1nm}$, and coagulation rates of each cluster to pre-existing particles were calculated from the Fuch's surface area for Atlanta, Boulder, Mexico City, and Hyytiälä (Kuang et al., 2010). Figure 3 shows how [$B_{eff}$] varies based on measured [$A_1$]. Each location exhibits clear differences in the range of [$B_{eff}$] regardless of measured sulfuric acid concentration. For example, Beijing shows the highest [$B_{eff}$] of any location with an average value of 2 pptv, indicating high concentrations of potent stabilizing compounds (e.g., DMA). The [$B_{eff}$] for Beijing are consistent with the measured [$B_{eff}$] of single-component injection of [DMA]~2-5 pptv (Fig. 1) which is similar to the measured [DMA]=2-3 pptv concentration at Beijing (Cai et al., 2021). In addition, the [$B_{eff}$] observed in Beijing contrasts with the other locations. Specifically, Hyytiälä Forest, where [$B_{eff}$]~0.02 pptv, is lower than even sulfuric acid-ammonia shown in Fig. 1. Mexico City and Atlanta are moderately polluted cities and exhibit [$B_{eff}$] of 0.8 and 0.1 pptv respectively. These values are less than Beijing but higher than Boulder and Hyytiälä Forest, suggesting that Mexico City and Atlanta contain moderate amounts and types of nucleating precursors.

The values of [$B_{eff}$] for all the sites except Beijing are lower than observed in the laboratory (Fig. 1 and 2). This could be due to uncertainties in calculating $J_{1nm}$ from >3 nm particle size distributions in Hyytiälä, Mexico City, Atlanta, and Boulder whereas $J_{1nm}$ was measured directly during the Beijing campaign. Beijing also exhibited the highest nucleation rates and condensation sink rates, while also having the lowest concentration of sulfuric acid. This means [$B_{eff}$] would need to increase to account for the higher nucleation rates with all other variables held constant. In addition, the lowest amine concentration examined in laboratory experiments for Fig. 1 and 2 was 1-2 pptv which may be higher than what occurred during the campaigns in Hyytiälä, Mexico City, Atlanta, and Boulder. Another reason the field [$B_{eff}$] are lower than observed in the laboratory is that other compounds exist in the atmosphere that help supress sulfuric acid nucleation. Further laboratory experiments are needed to better understand which and how specific compounds interfere with sulfuric acid nucleation.

Differences in temperature and relative humidity also play a role in [$B_{eff}$]. However, these differences may not be significant. A lower temperature should increase [$B_{eff}$] but Hyytiälä Forest (~0 °C) is lower than observed for Boulder (~22 °C). Boulder air quality is more impacted by agriculture (Flocke et al., 2020) and should contain more basic compounds which likely explains the higher [$B_{eff}$] compared to Hyytiälä Forest (Sipilä et al., 2015). This implies that the precursor compound concentration/composition plays a more significant role in [$B_{eff}$] than temperature. However, more experiments are needed to determine how [$B_{eff}$] is impacted by temperature and RH as this information is critical to predicting how [$B_{eff}$] varies around the world. Overall, these observations demonstrate that [$B_{eff}$] reflects the composition and concentration of stabilizing compounds detected in the atmosphere and can be used to model sulfuric acid nucleation rates in diverse areas.

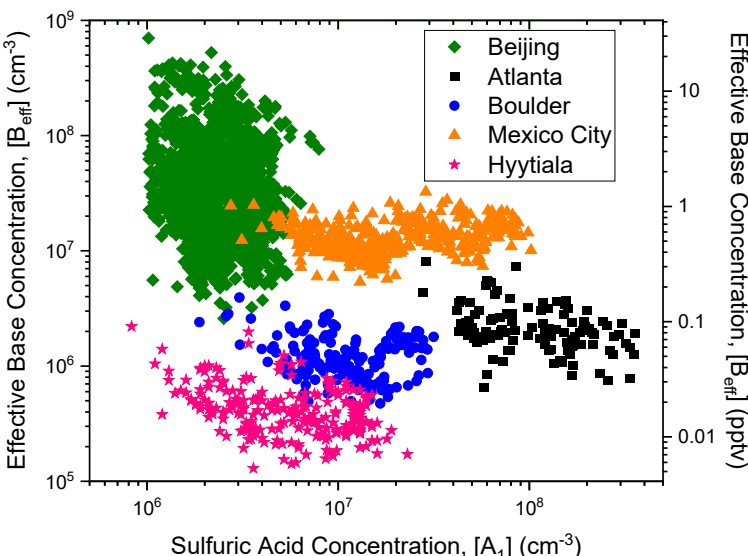

**Figure 3:** Comparison of the effective base concentration ([B$_{eff}$]) at various measured sulfuric acid concentrations ([A$_1$]) across five locations: green diamonds Beijing, China; red triangles Mexico City, Mexico; black squares Atlanta, Georgia; blue circles Boulder, Colorado; and pink stars Hyytiälä Forest, Finland.

Figure 4 compares [B$_{eff}$] to the weighted amine concentration ([DMA] + 0.2[TMA]) measured in Beijing (Cai et al.,
2021). In Figure 4, [B$_{eff}$] and the weighted amine concentration are positively correlated with a slope of 0.76 indicating that [B$_{eff}$] is sensitive to the amine concentration over a wide range of sulfuric acid concentrations. Furthermore, the data were divided into October, November, and December (2018) to explore how the seasons may affect precursor concentrations and nucleation rates. For October, more variation in [B$_{eff}$] is observed when compared to the weighted amine concentration. This variation could be due to weather and temperature changes that enhance or reduce sulfuric acid nucleation rates. Additionally,
other compounds likely exist in Beijing that nucleate with sulfuric acid which were not reported. November and December are significantly colder in Beijing, which would correlate with higher fuel (e.g., coal) burning and greater emissions of sulfuric acid and amines.

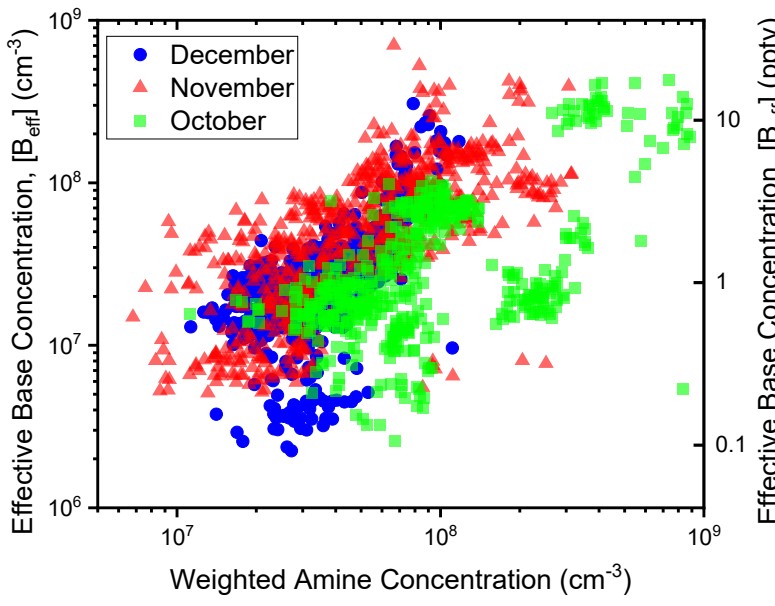

**Figure 4:** Comparison of effective base concentration from NPM ([B$_{eff}$]) with the weighted amine concentration measured in Beijing, China in 2018. October measurements are green squares, November orange triangles, and December blue circles.

## 4 Conclusion

240       The Nucleation Potential Model (NPM) is presented that simplifies predicting sulfuric acid nucleation rates in the complex atmosphere with two precursor concentrations: sulfuric acid and an effective base concentration ([B$_{eff}$]). The effective base concentration captures the amounts and types of stabilizing compounds that enhance sulfuric acid nucleation rates. NPM was applied to systems containing up to four atmospherically relevant bases reacting with sulfuric acid in a flow reactor. [B$_{eff}$] was determined from measured 1-nm particle concentrations, and its value depends heavily on the presence of strong stabilizing compounds, such as DMA and TMA, and their concentrations. [B$_{eff}$] values also reflect synergistic effects between

multiple compounds like DMA and ammonia. Finally, NPM was also used to calculate [B$_{eff}$] in various locations worldwide. Results show how the potency of the complex mixtures varies between polluted and unpolluted environments, and these observations did not require every potential stabilizing compound nucleating with sulfuric acid to be measured. [B$_{eff}$] can be determined from measured 1-nm particle concentrations produced from controlled reactions between a specified sulfuric acid concentration and a complex mixture. NPM complements current speciated measurements, such as those from a CIMS, by

providing additional insights into the potency of combined atmospheric compounds at enhancing sulfuric acid nucleation. Future field measurements will involve reacting atmospheric gases with a specific sulfuric acid concentration for a known amount of time to produce 1-nm particles to estimate [B$_{eff}$]. This will minimize possible interference with other particle formation mechanisms such as ion-induced or biogenic nucleation. NPM and further measurement of [B$_{eff}$] in diverse locations and seasons will help improve aerosol number concentrations predictions, reduce error in global climate models, and expand

understanding of the anthropogenic contribution to Earth's radiative balance.

**Data Availability:** Data is available upon request and is uploaded to The Index of Chamber Atmospheric Research in the United States (ICARUS) repository.

**Author Contribution:** JJ and CNJ both conceived of the model and experimental setup. JJ performed the experiments and data analysis. JJ and CNJ contributed to writing the paper.

**Competing Interests:** The authors declare that they have no conflict of interest

**Acknowledgments:** The authors thank Drs. Runlong Cai and Jingkun Jiang for providing their measurements from Beijing, China. In addition, the authors acknowledge the support from NSF AGS-1913504 and Aerosol Dynamics Inc. for lending the TSI 3789 vwCPC.

**Financial Support:** National Science Foundation AGS-1913504.

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
