# Peer review of "A Sulfuric Acid Nucleation Potential Model for the Atmosphere"

_Atmospheric Chemistry and Physics, 2022_

## Author Response (AR3)

To the Editor and Reviewers,

Thank you for the insightful comments regarding our manuscript entitled "A Sulfuric Acid Nucleation Potential Model for the Atmosphere." We appreciate the time and effort that the editor and reviewers have dedicated to providing valuable feedback on our manuscript. We have incorporated the changes suggested the reviewers.

Below is a point-by-point response to the reviewers' comments along with changes made in the manuscript colored in blue.

**Reviewer 1 Comments:**

**Reviewer 1 Summary:**

***This manuscript presents a novel idea on how to model atmospheric nucleation (formation rate of 1 nm particles) in environments dominated by acid-base nucleation mechanism. The theoretical approach behind the presented analysis appears scientifically sound. The paper is written in a relatively clear way, but it has some structural issues that need to be solved before I can recommend accepting this paper for publication. My main criticism in this regard is summarized below.***

We thank the reviewer for their insightful comments and have outlined below exactly how we made the paper clearer and more concise for the reader.

**Comment 1:**

***Section 2 should have separate sub-sections for the introduction of the NPM (current section 2.1) and for description of nucleation experiments (lines 96-106). After reading the paper, it remains somewhat unclear whether the data points presented in Figures 2 and 3 are taken from the experiments made earlier by the same research group in a series of papers cited on lines 97-102, or whether they are new experiments. This should be clarified in the paper. Furthermore, if the data are based on new experiment, the experimental method section should be expanded.***

We agree with the reviewer that sub-sections would be helpful for the reader in understanding the NPM and the nucleation experiments. We have added a section 2.2 to the paper to help separate model discussion from the experimental discussion. Additionally, we have changed the text to be much more explicit that these data were taken by our research group. We have also expanded upon the experimental methods section as suggested by the reviewer. See below for all highlighted changes related to this comment.

**Main Text Changes:**

[$B_{eff}$] was determined for nucleating systems consisting of sulfuric acid and various combinations of atmospherically relevant bases reacting in an extremely clean and repeatable flow reactor at 300 K and 20% relative humidity (RH). [$B_{eff}$] is likely influenced by temperature and RH. Lowering temperature would stabilize sulfuric acid clusters, leading to an increase in [$B_{eff}$] (Hanson and Eisele, 2002; Vehkamäki et al., 2002; Dunne et al., 2016). The effects of RH

are not clear and would depend on the concentration and composition of the other nucleation precursor vapors in the system (Olenius et al., 2017; Ball et al., 1999; Henschel et al., 2014; Merikanto et al., 2007). Future experiments will examine NPM over a wider range of temperature and RH to determine the impact this has on [$B_{eff}$]. The flow reactor system used for these measurements was constantly purged with a mixture of sulfuric acid, nitrogen, and water (Fomete et al., 2021; Jen et al., 2014; Ball et al., 1999). This creates extremely clean and repeatable conditions in the reactor. Baseline measurements are taken daily to verify the flow reactor's cleanliness and repeatability in concentration, temperature, and RH. The method for these baseline measurements is described in Fomete et al., 2021. [$A_1$]$_o$ and base concentration ([B]) were measured with a custom-built, transverse atmospheric pressure acetate/hydronium chemical ionization inlet coupled to a long time-of-flight mass spectrometer (Pittsburgh Cluster CIMS, PCC) (Fomete et al., 2021). The bases included dilute concentrations of ammonia ($NH_3$), methylamine (MA, $CH_3NH_2$), dimethylamine (DMA, $(CH_3)_2NH$), and trimethylamine (TMA, $C_3H_9N$) that are injected into the flow reactor by flowing nitrogen over a custom-made permeation tube (Fomete et al., 2021; Zollner et al., 2012).

**Comment 2:**

***Excluding the sentence on lines 178-179, section 3.2 is really not about application of NPM to the atmosphere, but rather about estimating (indirectly) the effective base concentration in a few environments where available observation allow such estimations. This should be reflected in the title of this sub-section.***

We agree with the reviewer and have changed section 3.2 title of the manuscript. See below for the main text changes.

**Main Text Changes:**

**3.2 Estimation of [$B_{eff}$] in Various Regions of the World**

**Comment 3:**

***Related to the previous comment, and for highlighting the value of this work, it would be essential to add some more text on how NPM could in practice be applied in different atmospheric environments (as compared to what is currently said on lines 178-179 and 201-203). Most importantly, how we can predict nucleation rates using NPM in environments where we do not have sophisticated measurements? Sulfuric acid concentrations can be predicted using models or proxies, but what would be a plan for predicting effective base concentrations besides comprehensive measurements?***

The reviewer brings up an excellent point. Most regions around the world do not have sophisticated precursor measurements. Unfortunately, current model predictions of sulfur dioxide (and therefore sulfuric acid) and ammonia (and methyl and diamine) already differ significantly from measured concentrations and contribute to the high uncertainty in predicting nucleation rates (Dunne et al., 2016; Yu and Luo, 2014; Ranjithkumar et al., 2021; Schiferl et al., 2016). Though our model simplifies the nucleation reactions into two measurable concentrations, the accuracy of NPM in predicting $J_{1nm}$ still depends heavily on high spatial and temporal

resolution measurements of sulfuric acid and [$B_{eff}$]. This is also the case for all other nucleation models. In addition, [$B_{eff}$] would ideally correlate with a compound currently included in model inventories, such as ammonia. However, our results from Figure 4 (Beijing observations) suggest [$B_{eff}$] correlates well with amine concentrations which are currently poorly represented in aerosol models.

In general, more measurements of nucleation precursors are needed regardless of nucleation model used. The strength of NPM is that it does not rely on challenging mass spectrometry measurements for precursor concentrations. [$B_{eff}$] can also be measured using a 1-nm CPC by reacting a known about of sulfuric acid with atmospheric air for a specific amount of time. The resulting concentration of 1-nm particles can be then used to calculated using NPM. The cost, size, and minimal power requirements of CPCs will allow researchers to easily measure [$B_{eff}$] at higher spatial and temporal resolution than is currently possible. We are currently preparing a manuscript detailing this instrument. In addition, [$B_{eff}$] can be estimated from measured 1-nm nucleation rates, as is done in the main text. An increasing number of groups are measuring $J_{1nm}$ (or $J_{1.7nm}$ which can be extrapolated to 1 nm with some degree of uncertainty) due to advances/availability in 1-nm particle instrumentation. Compiling the [$B_{eff}$] from all available $J_{1nm}$ measurements (with coagulation rates) will provide a more comprehensive picture on how [$B_{eff}$] depends on region to region and if there is a correlation between [$B_{eff}$] and an already modeled precursor concentration such as ammonia.

We have edited the main text to better emphasize the utility of NPM and how it will provide researchers with an easier way to measure precursor concentrations.

**Main Text Changes:**

This study demonstrates the dependencies of [$B_{eff}$] from a variety of stabilizing gas mixtures and how [$B_{eff}$] varies across diverse regions of the world.

The full impact of using the Nucleation Potential Model is two-fold: (1) The effective nucleation precursor concentration needed to predict 1-nm nucleation rates can be measured with a portable and cost-effective condensation particle counter (CPC), instead of a mass spectrometer. The increased development and deployment of 1-nm CPCs (Hering et al., 2017; Kuang et al., 2012; Lehtipalo et al., 2022) will enable researchers to measure [$B_{eff}$] at high spatial and temporal resolution which is currently challenging to achieve mass spectrometers. Furthermore, the combined observations from NPM with a CPC and mass spectrometry will also provide a detailed understanding on which compounds nucleate and the rate at which they nucleate. In addition, (2) the NPM is currently the only model that can represent nucleation of arbitrarily complex mixtures of compounds found in the atmosphere.

**Reviewer 2 Comments:**

**Reviewer 2 Summary:**

*This manuscript addresses the scientific question of understanding the drivers of new particle formation in different environments. This is important for understanding atmospheric aerosols from the perspective of both climate and air quality. This study has general implications for atmospheric science as the model developed for aerosol nucleation can be applied for different environments around the world, as demonstrated in the study, and is specifically useful for comparing different environments. I therefore consider this study to fit very well within the scope of ACP.*

*This paper presents a very novel tool, namely a new model for directly quantifying and comparing aerosol nucleation in different environments. The conclusions reached, that the model is useful for categorizing and comparing nucleation in different environments and can inform us about what precursors are involved in the nucleation process, are substantial.*

*The scientific methods and assumptions are valid and clearly outlined, with a few minor exceptions which I draw out in detailed comments below. The use of both flow-chamber an atmospheric data supports the conclusion that the model can inform us about the involvement of different precursors in nucleation, although some of the conclusions drawn about the unimportance of thermodynamic conditions and the amount of information the model gives us about precursors are not fully supported by the data presented. I have detailed below where this is the case.*

*Description of experiments and calculations lacks clarity in some areas, and this may hinder reproduction of results by fellow scientists. I have detailed where this occurs in my comments below. In addition, some important information on the method and how the model works has been relegated to the supplementary material, but seems essential enough to the understanding of the method that they might be better placed in the main text. The manuscript is generally well structured and the language is fluent and precise, except for the final section in the supplementary material, where I struggle to understand the main point as detailed in the specific comments below.*

*Authors mostly give proper credit to related work, but I have noticed a few missing citations, which I detail below. The title clearly reflects the contents of the paper and the abstract mostly provides a concise and complete summary. An issue I have throughout the study is a conflation between different precursor potency/amounts and thermodynamic conditions. Following more through addressing of this issue in the main manuscript, I believe this point should also be mentioned in the abstract.*

*This study is an interesting and important contribution to understanding and comparing aerosol nucleation over a variety of conditions. There are some open issues that need resolving/clarifying before publication*

We thank the reviewer for their detailed comments and suggestions on how to bring more clarity to our manuscript. We have thoroughly addressed all the comments outlined below which has led to many positive changes in our manuscript.

**Comments on the Introduction:**

**Comment 1:**

*Line 30 – discussion of sulfuric acid driven nucleation gives the impression that this only occurs in the presence of bases, even though organic acids, oxidized organics, ions and water mentioned later in the same paragraph. This may be a distinction the authors are making between pure binary nucleation (sulfuric acid and water only) and other types of nucleation involving sulfuric acid, but not exclusively. Clarity of the manuscript would be improved by addressing this explicitly.*

We agree with the reviewer that this sentence could be improved by providing more clarity to the reader. Below, we have revised the manuscript to reflect these changes.

**Main Text Changes:**

Aerosol nucleation in the troposphere is primarily driven by sulfuric acid (Kuang et al., 2008; Sihto et al., 2006; Sipilä et al., 2010; Lee et al., 2019; Weber et al., 1996a, 1997; Kulmala et al., 2004) which reacts with a large variety of compounds, to form particles (Kürten et al., 2016a; Glasoe et al., 2015; Weber et al., 1998; Kirkby et al., 2011; Jen et al., 2014; Coffman and Hegg, 1995; Almeida et al., 2013).

**Comment 2:**

*Line 60 – "most global climate models do not account for sulfuric acid nucleation" – my understanding is the models that include nucleation almost always account for pure binary, but not always nucleation involving additional species. Is this what the authors meant? Clarification and references are required here.*

We understand the reviewer's confusion with the previous phrasing and have updated it to be clearer. Below are the changes made to the manuscript.

**Main Text Changes:**

Currently, most global climate models only account for sulfuric acid binary or ternary nucleation with water or water and ammonia(Semeniuk and Dastoor, 2018). Only a few models incorporating power-law nucleation models (Gordon et al., 2017; Zhao et al., 2020; Dunne et al., 2016). However, experimental observations indicate that even low concentrations of other stabilizing compounds can enhance sulfuric acid nucleation rates beyond those predicted from models (Li et al., 2020; Wang et al., 2018).

**Comment 3:**

*Line 65 – "emissions inventories used in global climate models only contain SO$_2$ and ammonia" – it would be useful to reference the emissions inventories to which they are*

*referring here. Some inventories do include, for example, emissions of volatile organic compounds in the CEDS emissions database (Hoesly et al., 2018).*

We have included more sources and updated the text as shown below.

**Main Text Changes:**

However, experimental observations indicate that even low concentrations of other stabilizing compounds can enhance sulfuric acid nucleation rates beyond those predicted from models (Li et al., 2020; Wang et al., 2018). Moreover, many emission inventories used in global climate models only contain emission factors for sulfur dioxide and ammonia (Semeniuk and Dastoor, 2018; Lee et al., 2013; Dunne et al., 2016; Spracklen et al., 2008) with some including volatile organic compounds (Hoesly et al., 2018).

**Comment 4:**

*Line 67 – "these factors contribute to significant model error in predicting aerosol number concentrations in regions with no dominant nucleation pathway" – This needs supporting evidence. There are a number of existing studies comparing measured number concentrations to observations in diverse regions that should be cited here.*

While there are models that have accurately predicted particle nucleation rates in diverse regions, there are still many regions that have either overpredicted or underpredicted particle nucleation rates (Dunne et al., 2016; Kerminen et al., 2018; Ranjithkumar et al., 2021).

**Main Text Changes:**

Furthermore, only sparse measurements, both in time and space, exist of the numerous precursor compounds in the atmosphere. Combined, these factors contribute to significant model error in predicting aerosol number concentrations in regions with no dominant nucleation pathway (Dunne et al., 2016; Kerminen et al., 2018; Ranjithkumar et al., 2021).

**Comment 5:**

*Line 74 – "circumvents the need to deploy a mass spec" – only in the sense of capturing nucleation rates. More care should be taken to present it in this light, after all chemical info is needed also for predicting growth, hygroscopicity, emissions etc.*

We agree with the reviewer that it is also important to point out that this measurement technique complements mass spectrometry measurements. Mass spectrometers provided detailed chemical information but it is often difficult to know how specific compounds are nucleating based on observed mass spectra due to inherent uncertainty in chemically ionizing freshly formed clusters. We have edited manuscript and highlighted those changes below.

**Main Text Changes:**

This study demonstrates the dependencies of $[B_{eff}]$ from a variety of stabilizing gas mixtures and how $[B_{eff}]$ varies across diverse regions of the world.

The full impact of using the Nucleation Potential Model is two-fold: (1) The effective nucleation precursor concentration needed to predict 1-nm nucleation rates can be measured with a portable and cost-effective condensation particle counter (CPC), instead of a mass spectrometer. The increased development and deployment of 1-nm CPCs (Hering et al., 2017; Kuang et al., 2012; Lehtipalo et al., 2022) will enable researchers to measure [$B_{eff}$] at high spatial and temporal resolution which is currently challenging to achieve with mass spectrometers. Furthermore, the combined observations from NPM with a CPC and mass spectrometry will also provide a detailed understanding on which compounds nucleate and the rate at which they nucleate.

**Comment 6:**

*The method uses concentrations starting at 1 nm. Many atmospheric measurements start at higher diameters only, and measuring down to 1 nm can be extremely challenging in some environments, such as from aircraft. It would be helpful if the authors considered this issue and commented on whether this limits the use of the method in the introduction.*

We agree with the reviewer that many atmospheric measurements do not extend down to 1 nm. Many field-deployed CPCs have d50 cut-points around 3 nm. However, the technology for 1-nm CPCs is rapidly developing with more robust instruments entering the market. For example, Aerosol Dynamics Inc. (ADI) has recently developed a 1-nm water CPC which is sold commercially through TSI (Hering et al., 2017). ADI has also collaborated with Dr. Chongai Kuang to develop a DEG inlet for their rugged MAGIC CPC which extends the d50 to ~1 nm (Kuang, 2018). We are very confident that 1-nm CPCs will be much more widespread soon which will allow researchers to apply NPM in locations around the world.

**Main Text Changes:**

This study demonstrates the dependencies of [$B_{eff}$] from a variety of stabilizing gas mixtures and how [$B_{eff}$] varies across diverse regions of the world.

The full impact of using the Nucleation Potential Model is two-fold: (1) The effective nucleation precursor concentration needed to predict 1-nm nucleation rates can be measured with a portable and cost-effective condensation particle counter (CPC), instead of a mass spectrometer. The increased development and deployment of 1-nm CPCs (Hering et al., 2017; Lehtipalo et al., 2022; Kuang, 2018) will enable researchers to measure [$B_{eff}$] at high spatial and temporal resolution which is currently challenging to achieve with mass spectrometers.

**Comments on Section 2:**

**Comment 7:**

*How do the calculations of $B_{eff}$ detailed in this section apply to different temperatures and RHs? Confining the method to 300 K and 20 % RH seems to limit the global applicability.*

We agree with the reviewer that confining the temperature and RH to 300 K and 20% respectively limits global applicability. Previous nucleation kinetic experiments indicate that lower temperatures reduce sulfuric acid cluster evaporation rates (Hanson and Eisele, 2002; Vehkamäki et al., 2002; Dunne et al., 2016). This would be reflected as a higher [$B_{eff}$] in NPM.

The effects of relative humidity on nucleation rates are less obvious and depend highly on the chemical system. For example, increasing relative humidity for sulfuric-ammonia nucleating system has little to no impact on nucleation rates (Ball et al., 1999; Henschel et al., 2014; Merikanto et al., 2007), and relative humidity has minimal impact on nucleation rates sulfuric acid-DMA/TMA (i.e. potent nucleating agents) (Olenius et al., 2017). We have added clarifying language to the text to note this.

**Main Text Changes:**

[$B_{eff}$] was determined for nucleating systems consisting of sulfuric acid and various combinations of atmospherically relevant bases reacting in an extremely clean and repeatable flow reactor at 300 K and 20% relative humidity (RH). [$B_{eff}$] is likely influenced by temperature and RH. Lowering temperature would stabilize sulfuric acid clusters, leading to an increase in [$B_{eff}$] (Hanson and Eisele, 2002; Vehkamäki et al., 2002; Dunne et al., 2016). The effects of RH are not clear and would depend on the concentration and composition of the other nucleation precursor vapors in the system (Olenius et al., 2017; Ball et al., 1999; Henschel et al., 2014; Merikanto et al., 2007). Future experiments will examine NPM over a wider range of temperature and RH to determine the impact this has on [$B_{eff}$].

**Comment 8:**

*It is not clear to me from this section how losses tare taken into account. This is somewhat addressed in the supplementary material, but would be better addressed here to give the reader a full understanding of the method. Furthermore, the description of how losses are accounted for in the supplementary is unclear in places, as detailed below.*

NPM considers two types of losses: coagulation to larger clusters and wall loss. For the flow reactor experiments, coagulation to larger clusters is limited to clusters that have formed via nucleation as there are no pre-existing particles. Coagulation loss to small clusters is explicitly accounted for in the cluster balance equations (Equation S1). For examining [$B_{eff}$] from field-measured $J_{1nm}$, coagulation to pre-existing particles is also included (and wall losses are removed). We have modified the main text to clarify these points.

**Main Text Changes:**

The final step in Reaction 1 is the formation of the tetramer, $N_4$. At the tetramer size, the particles are approximately 1 nm in diameter or 1.3 nm in mobility diameter (Chen et al., 2012; Jen et al., 2015; Larriba et al., 2011). Coagulation losses are estimated from the collision rate constant between clusters. Any cluster formed through $N_8$ in size is accounted for in the total concentration of particles. Coagulation loss to larger particles (i.e., growth to sizes larger than $N_8$) is not included in this model when no pre-existing particles are present. Coagulation to pre-existing particles is included as a separate loss term when analyzing ambient observations. Cluster balance equations (i.e., rates laws) for Reaction 1 are provided in the supplementary information (SI, Equation S1).

**Comments on Section 3:**

**3.1:**

**Comment 9:**

*Flow reactor measurements are made at sulphuric acid concentrations of 1.4e8 cm$^{-3}$ and ammonia, DMA, MA, TMA concentrations between 0-35 pptv. It would be helpful to have a comment on how these relate to atmospherically relevant concentrations, and what the implications of this are for substantiating the method and the conclusions drawn from the flow reactor data.*

We agree with the reviewer that putting these numbers into context with atmospheric data will be helpful for the reader. See main text changes below.

**Main Text Changes:**

Figure 1 shows [$B_{eff}$] for the single component injections of $NH_3$, MA, DMA, and TMA in the sulfuric acid reactor. These atmospherically relevant compounds have previously been shown to nucleate with sulfuric acid at different rates (Jen et al., 2016, 2014; Kurtén et al., 2008; Glasoe et al., 2015). [$A_1$]$_o$, was measured daily and ranged between $9x10^7$ cm$^{-3}$ to $3x10^8$ cm$^{-3}$. Daily measurements of [$A_1$]$_o$ were then used as the initial concentration of sulfuric acid in the NPM. The average value for [$A_1$]$_o$ ([$A_1$]$_{o,avg}$=$1x10^8$ cm$^{-3}$) will be used for simplicity. While [$A_1$]$_o$ is higher than those typically measured in the atmosphere, any range of [$A_1$]$_o$ can be modelled as this parameter is an input to the NPM. Each base compound was injected at various measured [B], ranging from 0.5 to 32 pptv. The base concentrations examined in this study falls within the range observed in the atmosphere (Hanson et al., 2011; Cai et al., 2021; Kürten et al., 2016b).

**Comment 10:**

*Line 110 and Fig 1 – [$A_1$]$_0$ ~ 1.4e8 cm$^{-3}$, $t_{nucl}$~2s – Why are these approximate and what gives the range? Does this uncertainty affect the results in fig 1 at all? Given the scatter in the data in fig 1, uncertainties in [B] and [$B_{eff}$] appear noticeable on the scale of the graph and so should be shown. This also applies to fig 2.*

The approximate symbol on [$A_1$]$_o$ is due to small but measurable changes to sulfuric acid concentration on a day-to-day basis. Throughout the course of the experiments, [$A_1$]$_o$ ranged from $9x10^7$ cm$^{-3}$ to $3x10^8$ cm$^{-3}$, with an average value of $1.4x10^8$. To determine [$B_{eff}$], the [$A_1$]$_o$ used is the valued measured for that day's experiments, while the average [$A_1$]$_o$ value was used to provide the reader with approximate value of [$A_1$]$_o$. However, this may be confusing and misleading to reader. We have changed the text to explain [$A_1$]$_o$ more thoroughly. Additionally, we believe $1.4x10^8$ contains too many significant digits, due to the error of this value being around a factor of 2. As such, we have updated the average value to $1x10^8$ cm$^{-3}$. We have also added a subscript to [$A_1$]$_o$ to better clarify that this is just the average value across all sets of experiments.

$t_{nucl}$ is estimated from the centerline velocity of the flow reactor. The reactants experience a range of velocities within the laminar flow which provides a $t_{nucl}$ range between 2-27 s. However, the vWCPC samples from the centerline so most of the nucleation reactions occur at 2 s.  We

changed the nucleation time to be $t_{nucl}=2$ s since this value is constant across all sets of experiments.

**Main Text Changes:**

Figure 1 shows $[B_{eff}]$ for the single component injections of $NH_3$, MA, DMA, and TMA in the sulfuric acid reactor. These atmospherically relevant compounds have previously been shown to nucleate with sulfuric acid at different rates (Jen et al., 2016, 2014; Kurtén et al., 2008; Glasoe et al., 2015). $[A_1]_o$, was measured daily and ranged between $9x10^7$ cm$^{-3}$ to $3x10^8$ cm$^{-3}$. Daily measurements of $[A_1]_o$ were then used as the initial concentration of sulfuric acid in the NPM. The average value for $[A_1]_o$ ($[A_1]_{o,avg}=1x10^8$ cm$^{-3}$) will be used for simplicity.

**Comment 11:**

*Line 115 – "short nucleation time when compared to previous experimental studies" – this needs references, unclear which studies are being referred to.*

We agree with the reviewer. See below for updated text.

**Main Text Changes:**

This constant $[B_{eff}]$ trend suggests that $NH_3$ does not significantly stabilize sulfuric acid clusters and enhance nucleation rates under the experimental conditions in the flow tube. This is expected due to the relatively short nucleation time when compared to previous flow reactor studies (Jen et al., 2016; Glasoe et al., 2015). In contrast, $[B_{eff}]$ increases up to ~40 pptv with increasing [MA], demonstrating that this compound enhances sulfuric acid nucleation greater than $NH_3$.

**Comment 12:**

*Line 121 – Authors mention that nucleation rates plateau when coagulation rate are larger than predicted by model. Why is the model unable to deal with these coagulation rates? How does it deal with coagulation rates? This was not completely clear to me from the main manuscript and the description in the SM also left me with some questions (see below). Does this present a limit of applicability of the model e.g. can it work in polluted regions? Or plumes from volcanoes or aircraft?*

Currently, the NPM only captures coagulation of clusters between $N_1$ to $N_4$ to form clusters up to $N_8$ in size. The plateau in nucleation rates is caused when nucleation and growth rates are high, there is a much higher fraction of particles that are larger than $N_8$. Consequently, these large particles are increasing coagulation rates with freshly formed clusters beyond what the model is explicitly modeling. More accurate coagulation rates can be included in the model if the full particle size distribution were measured. Size distribution measurements at the sub-3 nm size, using a scanning mobility particle sizer, are very uncertain (Jiang et al., 2011). Consequently, we have designed our flow reactor experiments to prevent the formation of high concentration of large particles (>1 nm).

Unlike the flow reactor experiments, the atmosphere, both polluted and relatively clean environments, contains significant concentrations of preexisting particles (>3 nm) which can be

measured with higher certainty using a SMPS. The coagulation rate of precursors and freshly formed clusters can be estimated from the surface area of the pre-existing particle population (McMurry, 1983). We have included this scavenging loss term (i.e., coagulation loss to pre-existing particles) in NPM when examining field measurements.

Per the reviewer comment above (Comment #8), the main text was clarified to better explain how the model treats coagulation.

**Comment 13:**

*Line 140 "MA, NH3 and TMA concentrations do not change significantly" – is this assessment justified? NH3 concentration more than doubles, and TMA goes from nothing to 2 pptv – Other studies e.g. (Kurten et al., 2016) show that nucleation rates can be very sensitive to small concentrations of ammonia? Nucleation rates for TMA alone seems to have a strong response to the TMA concentration changes of less than 6 pptv in fig 1, and seems to have the strongest response at the lowest concentrations, indicating that 0-2 pptv may indeed be significant. In fig 2, the difference in J for DMA and DMA+73pptv NH3 is large enough to suggest that the difference of 25 pptv between the 2 and 3 component systems may indeed be significant. The ability of the reader to judge the significance of these concentration changes is also impaired by the lack of error bars.*

Figure 2 shows that $NH_3$ and MA, the measured base concentration does vary of up to 4 pptv for MA, and up to 60 pptv of $NH_3$ across the various mixture trials. However, from the results in Figure 1, an increase in MA and NH3 by these amounts would only increase [$B_{eff}$] by ~5 pptv. Adding MA to the DMA+NH3 only slightly increased [$B_{eff}$] which can be attributed to SA+MA nucleation.

Adding 2 pptv TMA to DMA+MA+NH3 did not appreciably increase [$B_{eff}$] beyond the scatter of the data points. There are likely a few different reasons why [$B_{eff}$] does not vary with the addition of each basic compound. (1) The 3 and 4-base component systems contain more strong nucleating base than sulfuric acid at low [DMA]. Thus, the addition of 1-2 pptv of TMA or addition of a significant amount of $NH_3$ will not impact particle nucleation rates due to sulfuric acid being the limiting reactant. (2) The coagulation rates become significant at higher concentrations of base and thus the model may not be capturing coagulation with larger particles. Coagulation rate explanation has been expanded upon in the main text more explicitly in Comment #8.

We agree with the reviewer that errors bars will help the reader determine how the various mixtures differ. We had added error bars to Figure 1 and 2 based upon fluctuations in particle concentrations during the measurement. Fluctuations in particle concentrations capture small variations in injected base concentrations and flow profiles.

**Main Text Change #1:**

Figure 2 also shows mixtures containing combinations of $NH_3$, MA, and TMA with varying amounts of DMA. Again, an increase in [DMA] leads to an increase in [$B_{eff}$], and all mixture curves display an enhancement to nucleation compared to pure sulfuric acid-DMA nucleation.

There is no significant distinction in the trends of [$B_{eff}$] between the 3 and 4 component mixture curves. This could be due to the higher base concentrations in these systems compared to the sulfuric acid concentration which results in particles being formed at the sulfuric acid collision limit. Addition of more bases could also help grow particles, causing higher coagulation losses not captured in the coagulation loss term in NPM. As discussed further in the SI, NPM only accounts for coagulation with particles up to of $N_8$ in size, indicating that NPM may not be capturing coagulation effects in the system saturated with base.

**Main Text Change #2:**

Regardless, observations from Fig. 2 indicate that NPM determines if a complex mixture of compounds will enhance sulfuric acid nucleation solely using measurements from the vwCPC.

The measured uncertainty in [$B_{eff}$] observed for the mixture experiments in Fig. 2 is higher than the single-component results (Fig. 1). The error bars were estimated from the standard deviation in the concentration of particles for each experiment. Fluctuations in particle concentrations capture the small variation in injected base concentrations and flow profiles. Additionally, the mixture experiments were measured over multiple days while many of the single component measurements were taken in 1-2 days. There are likely small day-to-day changes in the mixing within the dilution system which would increase the uncertainty across a longer time frame of measurements.

[Figure]

**Comment 14:**

*Line 140 – I'm not sure that the argument that NH3-DMA synergistic reaction dominates given small change in $B_{eff}$ with addition of MA and TMA is justified. NH3 concentrations vary a lot between the three systems.*

While NH$_3$ does vary between the three systems, Figure 1 shows that concentration differences between 15-73 pptv of NH$_3$ does not change [$B_{eff}$]. The 3-component and 4-component mixtures

are potentially showing the synergistic reactions of DMA and $NH_3$ with a linear addition of MA and TMA reactions with sulfuric acid.

**Main Text Changes:**

Additionally, [$B_{eff}$] is ~ 60 pptv at 10 pptv of DMA in the DMA and $NH_3$ curve in Fig. 2, while [$B_{eff}$] is ~100 pptv at 10 pptv of DMA for the 3 and 4 component mixture curves in Fig. 2. These observations imply that $NH_3$ and DMA are reacting synergistically with sulfuric acid, while MA and TMA are individually reacting with sulfuric acid to contribute the additional 40 pptv to [$B_{eff}$]. However, a computational chemistry model is required to draw further conclusions on how these molecules are reacting in a complex mixture. Regardless, observations from Fig. 2 indicate that NPM determines to what extent a complex mixture of compounds will enhance sulfuric acid nucleation solely using measurements from the vwCPC.

**Comment 15:**

*Line 145 – "NPM can determine HOW a complex mixture of compounds enhances SA nucleation" – this seems to be overstating the case. NPM can show that a complex mixture enhances the nucleation, and measure by how much, but cannot point to how (as in the chemical mechanism) this is done as is claimed above – or at least doesn't with the data available for the reasons mentioned in the comment on line 140.*

We agree with the reviewer that the way this was previously phrased was misleading to the reader of the intent for the model. Please see updated changes below.

**Main Text Changes:**

These observations imply that $NH_3$ and DMA are reacting synergistically with sulfuric acid, while MA and TMA are individually reacting with sulfuric acid to contribute the additional 40 pptv to [$B_{eff}$]. However, a computational chemistry model is required to draw further conclusions on how these molecules are reacting in a complex mixture. Regardless, observations from Fig. 2 indicate that NPM determines to what extent a complex mixture of compounds will enhance sulfuric acid nucleation solely using measurements from the vwCPC.

**Comment 16:**

*Line 150 – The uncertainty mentioned here really needs to be illustrated on the figure somewhere to help the reader see whether or not it impacts the trends.*

Uncertainty in [$B_{eff}$] has been determined by using the standard deviation in particle concentration measurements for the vwCPC. We have added error bars to Figure 1 and 2 based upon fluctuations in particle concentrations during the measurement. As mentioned in comment 13, fluctuations in particle concentrations capture small variations in injected base concentrations and flow profiles.

**Main Text Changes:**

While [$A_1$]$_o$ is higher than those typically measured in the atmosphere, any range of [$A_1$]$_o$ can be modelled as this parameter is an input to the NPM. Each base compound was injected at various measured [B], ranging from 0.5 to 32 pptv. The base concentrations examined in this study falls

within the range observed in the atmosphere (Hanson et al., 2011; Cai et al., 2021; Kürten et al., 2016b). Note, the error bars in Fig. 1 represent how the standard deviation in particle concentration measurements effects [$B_{eff}$].

**3.2:**

**Comment 17:**

*Line 175 – I question the claim that temperature and RH effects may not be significant given similar $B_{effs}$ in Boulder and Hyytiälä. Since temperature and RH are very different for these two locations, could not the similar $B_{eff}$ because by compensating effects from differing precursors and thermodynamic conditions? This should be highlighted also in the introduction to the NPM as it is important that future users of the NPM fully understand that the model puts these influences together. This does not detract from the usefulness of the model, especially to compare observations where T and RH are known, but does need to be properly acknowledged in analyses such as this. It could be a useful extension of the future work proposed in the next line to measure $B_{eff}$ for known H2SO4 concentrations to also map $B_{eff}$ for known H2SO4 concentrations over a range of atmospherically relevant temperatures and RHs.*

We agree with the reviewer that T, RH, and different mixture of precursors could have competing effects on the observed [$B_{eff}$] for these different locations. We have modified the text to clarify that future studies need to examine the dependencies of T and RH on [$B_{eff}$]. Additional text was also included in the introduction to highlight T and RH effects as shown in Comment #7.

**Main Text Changes:**

Differences in temperature and relative humidity also play a role in [$B_{eff}$]. However, these differences may not be significant. A lower temperature should increase [$B_{eff}$] but Hyytiälä Forest (~0 °C) is lower than observed for Boulder (~22 °C). Boulder air quality is more impacted by agriculture  (Flocke et al., 2020) and should contain more basic compounds which likely explains the higher [$B_{eff}$] compared to Hyytiälä Forest(Sipilä et al., 2015). This implies that the precursor compound concentration/composition plays a more significant role in [$B_{eff}$] than temperature. However, more experiments are needed to determine how [$B_{eff}$] is impacted by temperature and RH as this information is critical to predicting how [$B_{eff}$] varies around the world.

**Comment 18:**

*Fig 3 - this clear distinction between the different environments is really powerful and such a strong endorsement of the potential power of this mode. It is very interesting also to see how, in contract to other locations in this study $B_{eff}$ in Beijing is mostly controlled by factors other than sulfuric acid concentration.*

Yes, we agree with the reviewer that it is interesting to see how [$B_{eff}$] can depend on many other factors including condensation sink and mixture of precursor compounds. More measurements of

[B$_{eff}$] in diverse environments will provide a more comprehensive picture on why [B$_{eff}$] varies due to changes in precursors, T, RH, and other regional factors. In addition, we have improved the coagulation loss rates to pre-existing particles in all the campaigns by including more accurate A$_{fuchs}$ in NPM from each field campaign using measured particle size distributions. We have also updated the coagulation sink term in the steady-state model to account for changes in the size of each cluster that is lost to pre-existing particles. The updated Figure 3 shows [B$_{eff}$] is independent of sulfuric acid concentration which should be true given sulfuric acid concentration is an input into NPM. The discussion was expanded upon for this calculation as well as shown in comment #22.

**Main Text Changes:**

The equations of the NPM (Equation S1) were solved at steady state to determine [B$_{eff}$] from the observed J$_{1nm}$, and coagulation rates of each cluster to pre-existing particles were calculated from the Fuch's surface area for Atlanta, Boulder, Mexico City, and Hyytiälä (Kuang et al., 2010). Figure 3 shows how [B$_{eff}$] varies based on measured [A$_1$]. Each location exhibits clear differences in the range of [B$_{eff}$] regardless of measured sulfuric acid concentration. For example, Beijing shows the highest [B$_{eff}$] of any location with an average value of 2 pptv, indicating high concentrations of potent stabilizing compounds (e.g., DMA). The [B$_{eff}$] for Beijing are consistent with the measured [B$_{eff}$] of single-component injection of [DMA]~2-5 pptv (Fig. 1) which is similar to the measured [DMA]=2-3 pptv concentration at Beijing (Cai et al., 2021). In addition, the [B$_{eff}$] observed in Beijing contrasts with the other locations. Specifically, Hyytiälä Forest, where [B$_{eff}$]~0.02 pptv, is lower than even sulfuric acid-ammonia shown in Fig. 1. Mexico City and Atlanta are moderately polluted cities and exhibit [B$_{eff}$] of 0.8 and 0.1 pptv respectively. These values are less than Beijing but higher than Boulder and Hyytiälä Forest, suggesting that Mexico City and Atlanta contain moderate amounts and types of nucleating precursors.

The values of [B$_{eff}$] for all the sites except Beijing are lower than observed in the laboratory (Fig. 1 and 2). This could be due to uncertainties in calculating J$_{1nm}$ from >3 nm particle size distributions in Hyytiälä, Mexico City, Atlanta, and Boulder whereas J$_{1nm}$ was measured directly during the Beijing campaign. Beijing also exhibited the highest nucleation rates and condensation sink rates, while also having the lowest concentration of sulfuric acid. This means [B$_{eff}$] would need to increase to account for the higher nucleation rates with all other variables held constant. In addition, the lowest amine concentration examined in laboratory experiments for Fig. 1 and 2 was 1-2 pptv which may be higher than what occurred during the campaigns in Hyytiälä, Mexico City, Atlanta, and Boulder. Another reason the field [B$_{eff}$] are lower than observed in the laboratory is that other compounds exist in the atmosphere that help supress sulfuric acid nucleation. Further laboratory experiments are needed to better understand which and how specific compounds interfere with sulfuric acid nucleation.

[Figure]

**Comment 19:**

*Fig 4 – overlap of opaque symbols hides some of the spread of the October and November data.*

We thank the reviewer for noticing this. We have made some symbols more transparent so the spread of Oct and Nov points can be better seen.

**Main Text Changes:**

[Figure]

**Conclusion:**

**Comment 20:**

*Line 195 – Again the possible variation of $B_{eff}$ due to temperature or RH is ignored. Is this assumed to be small compare with precursor concentrations? If so, are there data from controlled or measured conditions to illustrate this?*

We agree with the reviewer that T and RH could be influencing [$B_{eff}$], and have addressed these limitations in comment #7 and #17.

**Supplementary:**

**Comment 21:**

*Line 16 – is it reasonable to set cluster balances up to [N3] equal to zero? What is the justification fort this?*

The justification for setting all cluster balances up to [$N_4$] largely stems from the fact that the field data was taken under steady state conditions (Chen et al., 2012; Kerminen et al., 2018). Because of this, we use the steady-state form of the NPM as there is no longer a dependency on nucleation reaction time.

**Main Text Changes:**

For the steady-state case of the model, which was applied to atmospheric data, cluster balances through [$N_4$] are set equal to zero and the equation for $N_{>4}$ is removed (Chen et al., 2012; Kerminen et al., 2018; Kuang et al., 2008). This is because nucleation reactions in the field are calculated at the peak sulfuric acid concentration, which occurs when the $J_{1nm}$ has plateaued at large reaction times (i.e. steady state) (Chen et al., 2012; Kerminen et al., 2018). $J_{1nm}$ is calculated from the formation rate of [$N_4$].

**Comment 22:**

*Line 16 – Why does coagulation with larger clusters replace wall loss rates for atmospheric measurements? I understand that of course wall losses now don't exist, but coagulation with pre-existing particles also existed in the flow chamber. Or is this not taken into account in the chamber version? And if not, why not? Previous studies have shown that in chamber conditions self-coagulation can be an important loss impacting the calculation of nucleation rates significantly (Kurten et al., 2015).*

Coagulation with pre-existing particles in the flow reactor is largely non-existent because the flow reactor is purged with particle-free flows of nitrogen, humidified nitrogen, and sulfuric acid vapor. Daily baseline measurements with the vWCPC also indicated that there are negligible amounts of background particles before each nucleation experiment (Fomete et al., 2021). The flow reactor we use has a much shorter nucleation reaction time (2 s) compared to chambers experiments (hours) which prevents the formation of large particles. Thus, we do not include coagulation losses to pre-existing particles in the flow reactor or particles larger than $N_8$. We believe the main confusion here comes from our language of using "replaced" when we more meant that wall-losses are removed and coagulation rates to pre-existing particles are included.

**Main Text Changes:**

For the steady-state case of the model, which was applied to atmospheric data, cluster balances up to $[N_4]$ are set equal to zero (Chen et al., 2012; Kerminen et al., 2018; Kuang et al., 2008). This is because nucleation reactions in the field are calculated at the peak sulfuric acid concentration, which occurs when the $J_{1nm}$ has plateaued at large reaction times (i.e. steady state) (Chen et al., 2012; Kerminen et al., 2018). $J_{1nm}$ is calculated from the formation rate of $[N_4]$. Additionally, wall loss rates are removed from the model, while coagulation loss rates to pre-existing particles are added. The coagulation loss rate was calculated from the Fuch's surface area ($A_{fuchs}$) of measured particle size distributions (Kuang et al., 2010) during the various field campaigns. The condensation sink (CS) is calculated from $CS = \frac{1}{4}\bar{c} \cdot A_{fuchs}$ where $\bar{c}$ is the mean thermal speed. $A_{fuchs}$ is estimated for each $J_{1nm}$ value from the campaign by binning of $A_{fuchs}$ measurements based on the peak sulfuric acid concentrations for each field-campaign (Sihto et al., 2006; Iida et al., 2008; McMurry and Eisele, 2005; Cai et al., 2021; Eisele et al., 2006).

**Comment 23:**

*Line 18 – coagulation loss rate assumed constant for field campaigns – why was this done? And was it calculated from the measured size distributions? Did these change? And if so, which size distribution measurement was used? More detail needed here for others to be able to reproduce the method on other field campaign data.*

The averaged coagulation loss rates were found in Kuang et al., 2010. However, we are using measured size distributions to determine the coagulation rate loss from the various field campaigns. Coagulation loss values were then binned based on sulfuric acid concentration. And while coagulation loss is not solely dependent on sulfuric acid concentration, it is a good indicator for the coagulation losses (Kuang et al., 2010). In the binned version, the data shows less of a dependence on sulfuric acid concentration. This is expected as sulfuric acid concentration is an input into NPM. We have updated the figure and included an explanation of how we determined coagulation rate losses to pre-existing particles. See response to comment #18.

**Comment 24:**

*Line 21 – "first set of bars" unclear – refer to how they are labeled in the figure. This whole sentence is a bit unclear – is the point being made just that there are many particles sized between 1 and 2 nm? If so, this might be clearer with a linear y axis in fig S1 and the text could use some clarifying.*

We agree with the reviewer that this sentence was unclear. The text has been changed to clarify this.

**Main Text Changes:**

As seen in Figure S1, the left-hand bars at a high concentration of sulfuric acid ($[A_1]_o = 5\text{x}10^9$ cm$^{-3}$) show a high concentration of particles of $2\text{x}10^5$ cm$^{-3}$ at the 1-nm 50% cut-point ($d_{50}$), ($T_{conditioner} = 1$ °C, $T_{initiator} = 99$ °C), with almost half of these particles also present when the cut-point was changed to 2 nm ($T_{conditioner} = 2$ °C, $T_{initiator} = 90$ °C). This indicates that there is a

significant concentration of particles larger than 2 nm at high $[A_1]_o$ which would lead to inaccuracies in coagulation rates up to $N_8$ within NPM. The right-hand set of bars at lower $[A_1]_o$ shows a significantly lower particle concentration of $8\times10^3$ cm$^{-3}$ at $d_{50}=1$ nm and 30 cm$^{-3}$ at $d_{50}=2$ nm. Low concentrations of 2-nm particles suggests that most formed particles are less than 2 nm in diameter.

**Comment 25:**

*Line 27 – "This inferred size distribution is more compatible with the nucleation model, which accounts for particles up to N8 (larger than 1 nm)." Does this mean that the nucleation potential model does not work for size distributions dominated by smaller particles/clusters? Isn't this quite a lot of nucleation events and wouldn't that be a major limitation of the model? If that is the case, why is this not highlighted in the main manuscript*

The main limitation here resides in that the model is only accounting for the kinetic reactions of particles up to $N_8$. Meaning, that any interaction (formation, coagulation, and wall loss) of particles smaller than $N_8$ is captured in the NPM, but reactions occurring at larger sizes are not included NPM unless the coagulation rate with pre-existing particles is known. Particles larger than $N_8$ are no longer nucleating and are in the growth phase. Because of this, we do not consider it to be a major model limitation since the model is focused on capturing the nucleation process. We have changed the text below to try to make this point clearer to the reader.

**Main Text Changes:**

Low concentrations of 2-nm particles suggests that most formed particles are less than 2 nm in diameter. This inferred size distribution is compatible with the nucleation model, which explicitly accounts for the formation and loss rates for particles up to $N_8$ (larger than 1 nm). This method of varying $d_{50}$ cut-point to determine the size of the particles was used instead of a Scanning Mobility Particle Sizers (SMPS) due to the large uncertainty associated with charging particles in the 1-nm size range (Jen et al., 2015; Jiang et al., 2011).

**Comment 26:**

*The lack of SMPS scans for the flow tube – does this mean there are no size distribution measurements? And if so, how is the coagulation sink accounted for?*

Though we are able to measure the particle size distribution within the reactor, we have chosen not to use these results due to the enormous uncertainty in charging 1-2 nm particles (Jen et al., 2015; Jiang et al., 2011). More accurate size distributions, though with larger diameter bins, were obtained by changing the d50 of the vWCPC as shown in Figure S1. These results indicate that particles remain very small, around 1 nm geometric diameter, which is accounted for in NPM.

**Main Text Changes:**

This inferred size distribution is compatible with the nucleation model, which explicitly accounts for the formation and loss rates for particles up to $N_8$ (larger than 1 nm). This method of varying $d_{50}$ cut-point to determine the size of the particles was used instead of a Scanning Mobility Particle Sizers (SMPS) due to the large uncertainty associated with charging particles in the 1-

nm size range (Jen et al., 2015; Jiang et al., 2011). Future work will explore size-resolved measurements in greater detail to further increase the accuracy of NPM in estimating coagulation loss rates.

**Reviewer 3 Summary:**

*A new parameter Beff was proposed in this manuscript to depict the enhancing efficiency of atmospheric bases (especially amines) on sulfuric acid-driven nucleation. Derived from a simplified kinetic model and validated by flow tube experiments, the Nucleation Potential Model (NPM) was established and used to explain the NPF observations around the world. The manuscript is generally well written and organized. I recommend the publication of this paper after the following points are addressed.*

We thank the reviewer for their comments and have outlined how we addressed each comment below.

**Reviewer 3 Comments:**

**Comment 1:**

*As stated in lines 41-47, power-law nucleation models employ empirically derived parameters and only depend on a few nucleation precursors. Thus, they predict inaccurate nucleation rates. In fact, for the typical power-law nucleation model (like J = K[H2SO4]1-2), enhancing potential and amounts of base precursors can also be captured by the factor K, similar to Beff in NPM described in the manuscript. The main difference between this power-law nucleation model and NPM should be whether specific cluster kinetics are considered. Since many assumptions are made for cluster kinetics in NPM, it might be nice for the authors to clarify how NPM (Beff-H2SO4) performs better than the power-law nucleation model (K-H2SO4) in differentiating the enhancing potential of various bases and revealing different patterns for field measurements. This might be important for the further promotion of NPM.*

The reviewer brings up a specific example of a power-law nucleation model where the nucleation rate depends on the squared sulfuric acid concentration. This was first fitted from field observations presented in Weber et al., 1996 and it was hypothesized that nucleation rates were limited by the collisions of sulfuric acid. The pre-factor k<0.5 scales the nucleation rate as a fraction of the sulfuric acid collision-controlled rate limit. Some nucleating systems have been shown to have higher than squared dependency on sulfuric acid concentration; for example, sulfuric acid, ammonia, and water have exhibited powers between 3-4 (Dunne et al., 2016; Glasoe et al., 2015). The power on the sulfuric acid concentration drops to 2.5-3.7 for sulfuric acid and dimethylamine (Dunne et al., 2016; Glasoe et al., 2015). As with other power-law models, both the coefficient and the exponentials are fitted from observation data. The fitted parameters may indicate key rate-limiting steps (Sihto et al., 2006) or may have limited meaning (Kupiainen-Määttä et al., 2014). In contrast, NPM does not directly fit the power dependency but, as the reviewer notes, assumes the reaction kinetic scheme. Consequently, NPM does use a parameterized effective base concentration which is a gauge of how potent the compound or

mixture of compounds are at enhancing sulfuric acid cluster formation (nucleation rate). We have changed the main text to make this point clearer.

**Main Text Changes:**

These power-law models have been used to predict nucleation rates in areas such as Asian megacities, the Amazon Rainforest, and globally (Yao et al., 2018; Zhao et al., 2020; Dunne et al., 2016). The fitted coefficient and exponentials on the precursor concentration may be indicative of key rate limiting steps (Sihto et al., 2006) or may have no physical meaning (Kupiainen-Määttä et al., 2014). Furthermore, the power-law models are typically only dependent on two to three nucleation precursor concentrations, and thus cannot accurately predict nucleation rates in areas where numerous and unknown compounds are nucleating with sulfuric acid (Zhao et al., 2020).

**Comment 2:**

*In deriving the birth-death equilibrium equations (Equation S1), I suppose a coefficient of 0.5 should be multiplied for the source/sink items from identical clusters (i.e. k[N1]2, k[N3]2 and k[N4]2) to avoid overestimation of cluster collisions?*

For the birth-death equations, we have decided to instead include a coefficient of 2 in the loss-terms of identical clusters to account for the overestimation of the cluster collisions.

For example,

A1+A1 →A2

Formation rate of A2= ½*consumption rate of A1

Or

2*Formation rate of A2= consumption rate of A1

**Comment 3:**

*The parameter Beff literally represents the enhancing potential of precursors in sulfuric acid-driven nucleation. In fact, sulfuric acid is not always essential in all atmospheric nucleation events. For example, pure biogenic particles can also be formed by low-volatility vapors via neutral/ion-induced organic nucleation. Considering that the measured 1-nm particle concentrations are the sum of all sources, this uncertainty from the non-sulfuric acid part should at least be mentioned.*

We agree with the reviewer that there are other sources of particle formation in the atmosphere besides sulfuric acid nucleation; other compounds that could nucleate without sulfuric acid include iodocompounds, HOMS, and methanesulfonic acid. However, the field campaigns in this study specifically examined at nucleation events where sulfuric acid was present, implying that a large majority of the particles measured were from the sulfuric acid nucleation event. Other non-sulfuric acid pathways exhibit nucleation rates much slower than sulfuric acid-ammonia rates (Kirkby et al., 2016, 2011) at 273-300K. In addition, ion production at ground level is slow and

thus ion-induced nucleation pathways are slow compared to neutral sulfuric acid nucleation pathways (Kirkby et al., 2016, 2011; Lovejoy et al., 2004). Furthermore, we have described in the main paper that to measure [$B_{eff}$], a specific (and high, ~$10^8$ cm$^{-3}$) concentration of sulfuric acid is reacted with air for a known amount of time to produce 1-nm particles. This controlled formation of 1-nm particles with sulfuric acid will also dominate the nucleation pathways compared to non-sulfuric acid pathways.

**Main Text Changes:**

NPM complements current speciated measurements, such as those from a CIMS, by providing additional insights into the potency of combined atmospheric compounds at enhancing sulfuric acid nucleation. Future field measurements will involve reacting atmospheric gases with a specific sulfuric acid concentration for a known amount of time to produce 1-nm particles to estimate [$B_{eff}$]. This will minimize possible interference with other particle formation mechanisms such as ion-induced or biogenic nucleation. NPM and further measurement of [$B_{eff}$] in diverse locations and seasons will help improve aerosol number concentrations predictions, reduce error in global climate models, and expand understanding of the anthropogenic contribution to Earth's radiative balance.

**Comment 4:**

*Typical atmospheric concentration of NH3 is likely 2-3 orders of magnitude higher than that of amines, which is not the case in the presented flow reactor experiments. The impact of this deviation on conclusions should be considered. In addition, the authors should explain why DMA concentration is selected to represent the bases in multi-base experiments (Perhaps, DMA is the primary base species in enhancing sulfuric driven nucleation from field evidence).*

We agree with the reviewer that a more atmospherically relevant mixture of bases would contain a much higher concentration of NH$_3$. However, due to the very minimal change in [$B_{eff}$] across the range of NH$_3$ tested, it is believed that NH$_3$ has minimal impact on [$B_{eff}$] at such a short reaction time. Additionally, NH$_3$ concentrations are significantly higher than the sulfuric acid concentration in the reactor, similar to what is observed in the atmosphere. Future experiments will include even more complex mixtures of atmospherically relevant bases and organic acids to validate the NPM further.

To the second point, DMA was chosen as the primary base because of its known potency (i.e., near zero cluster evaporation rates) in reacting with sulfuric acid to form particles. The reaction kinetics of DMA+sulfuric acid have also been widely studied. The trends from these experiments demonstrate how [$B_{eff}$] primarily depends on the concentration of potent nucleation compounds (DMA, TMA). Additionally, DMA has been measured in several locations worldwide (Cai et al., 2021; Zhao et al., 2011; Kürten et al., 2016b; Almeida et al., 2013; Freshour et al., 2014).

**Several changes were made to the main text to correct typos and these changes are included in the tracked changes document.**

**References cited in response to reviewers:**

[revised manuscript text omitted]